Growth and physiological responses of submerged plant Vallisneria natans to water column ammonia nitrogen and sediment copper

Zhu Zhengjie 1 2
Song Siyuan 1 2
Li Pengshan 1 2
Jeelani Nasreen 1 2
Wang Penghe 1 2
Yuan Hezhong 2 3
Zhang Jinghan 2
An Shuqing 1 2
Leng Xin 1 2 lengx@nju.edu.cn
1 School of Life Science and Institute of Wetland Ecology, Nanjing University , Nanjing Jiangsu , PR China
2 Nanjing University Ecology Research Institute of Changshu , Changshu Jiangsu , PR China
3 School of Environmental Science and Engineering, Nanjing University of Information Science and Technology , Nanjing Jiangsu , PR China
Gupta Vijai Kumar
Electronic publication date: 2016 Apr 21
Publication date: 2016
Volume: 4
Electronic Location ID: e1953
Received 2016 Feb 16; Accepted 2016 Mar 29
Copyright: ©2016 Zhu et al.
Copyright year: 2016
Copyright holder: Zhu et al.
License: This is an open access article distributed under the terms of the Creative Commons Attribution License, which permits unrestricted use, distribution, reproduction and adaptation in any medium and for any purpose provided that it is properly attributed. For attribution, the original author(s), title, publication source (PeerJ) and either DOI or URL of the article must be cited.
License URL: https://creativecommons.org/licenses/by/4.0/

Keywords: Ammonia nitrogen, Sediment copper, Submerged plant, Vallisneria natans

Funding: Major Science and Technology Program for Water Pollution Control and Treatment 2012ZX07204-004 2015ZX07204-002 This study was funded by the Major Science and Technology Program for Water Pollution Control and Treatment (2012ZX07204-004 & 2015ZX07204-002). The funders had no role in study design, data collection and analysis, decision to publish, or preparation of the manuscript.

==============================
Background. The decline of submerged plant populations due to high heavy metal (e.g., Cu) levels in sediments and ammonia nitrogen (ammonia-N) accumulation in the freshwater column has become a significant global problem. Previous studies have evaluated the effect of ammonia-N on submerged macrophytes, but few have focused on the influence of sediment Cu on submerged macrophytes and their combined effects.

Methods. In this paper, we selected three levels of ammonia-N (0, 3, and 6 mg L−1) and sediment Cu (25.75 ± 6.02 as the control, 125.75 ± 6.02, and 225.75 ± 6.02 mg kg−1), to investigate the influence of sediment Cu and ammonia-N on submerged Vallisneria natans. We measured the relative growth rate (RGR), above- and below- ground biomass, chlorophyll, non-protein thiol (NP-SH), and free proline.

Results and Discussion. The below-ground biomass of V. natans decreased with increasing Cu sediment levels, suggesting that excessive sediment Cu can result in significant damage to the root of V. natans. Similarly, the above-ground biomass significantly decreased with increasing ammonia-N concentrations, indicating that excessive water ammonia-N can cause significant toxicity to the leaf of V. natans. In addition, high ammonia-N levels place a greater stress on submerged plants than sediment Cu, which is indicated by the decline of RGR and chlorophyll, and the increase of (NP-SH) and free proline. Furthermore, high sediment Cu causes ammonia-N to impose greater injury on submerged plants, and higher sediment Cu levels (Cu ≥ 125.75 mg kg−1) led to the tolerant values of ammonia-N for V. natans decreasing from 6 to 3 mg L−1. This study suggests that high sediment Cu restricts the growth of plants and intensifies ammonia-N damage to V. natans.

Introduction

Rapid worldwide economic and industrial development has resulted in eutrophication and heavy metal pollution of freshwater bodies, which has subsequently led to the deterioration of these aquatic environments (Cheung et al., 2003; Zhang et al., 2012). Vegetation restoration has emerged as an effective way of improving water quality by reducing eutrophication and removing heavy metals from soil and wastewater, because macrophytes have tremendous capacity of absorbing nutrients and toxic metals from polluted soil and water (Gupta & Chandra, 1998; Meagher, 2000; Ali, Bernal & Ater, 2004; Nixon, 2009). However, large numbers of pioneer plants have been blindly planted for ecological restoration. Successful ecological restoration depends on planting submerged macrophytes below the tolerant levels of water ammonia-N and sediment-Cu.

At low levels, copper (Cu) is an essential trace element for a variety of cells and tissues in submerged plants, but at high concentrations, it can cause phytotoxicity (Ali, Bernal & Ater, 2002; Ali, Bernal & Ater, 2004; Ali et al., 2015). Excessive Cu accumulation results in detrimental effects on several physiological and biochemical processes in plants and can also inhibit growth (Påhlsson, 1989; Fernandes & Henriques, 1991; Wang et al., 2013). Moreover, heavy metals such as Cu are generally bound to particulate matter and eventually become incorporated into sediments rather than water columns (Wang et al., 2010b; Ng, 2015). Taking into account that previous studies generally focused on water column copper, the effects of excessive sediment Cu on macrophytes such as Vallisneria natans (V. natans) need to be investigated.

Ammonia-N is an important nutrient source to submerged macrophytes at low concentrations, but it can be toxic at higher levels (Best, 1980; Smolders, Van Riel & Roelofs, 2000; Cao et al., 2007; Ellis, Craft & Stanford, 2015). Damaging concentrations of ammonia-N can inhibit photosynthesis, trigger oxidative stress, and cause water loss in plants (Smolders, Van Riel & Roelofs, 2000; Cao, Ni & Xie, 2004; Li, Cao & Ni, 2007; Neuberg et al., 2010). Furthermore, the toxicity of ammonia-N in water bodies was significantly influenced by high contents of Cu because Cu caused the accumulation of excess NH4+ in the cytosol (Llorens et al., 2000; Mazen, 2004). However, although previous studies have commonly evaluated the effect of ammonia-N on submerged macrophytes, few have focused on the combined effects of ammonia-N and sediment Cu.

As seen in other polluted freshwater bodies, the ecosystem of the Huai River in China has been severely degraded by excessive pollutant discharge that produced high heavy metal contents, including Cu, and excessive ammonia-N levels in the water column (Zhang et al., 2010; Xia et al., 2011; Yuan et al., 2015a). The Huai River reportedly has ammonia-N concentrations up to 29.70 mg L−1, and sediment Cu concentrations up to 208.8 mg kg−1 (Ren et al., 2015; Yuan et al., 2015b). In this study, V. natans, a ubiquitous submerged plant in the Huai River, was selected as the treatment subject. Changes in plant growth and distribution (Britto & Kronzucker, 2002; Xie, An & Wu, 2005) and fluctuations of many metabolites such as chlorophyll (Prasad et al., 2001), non-protein thiol (NP-SH) (Maserti et al., 2005), and proline (Ashraf & Foolad, 2007) are important indexes to measure the response of a plant to environmental stress. Thus, the relative growth rate (RGR), above- and below-ground biomasses, chlorophyll, NP-SH, and free proline were selected to test (1) growth response of V. natans to sediment Cu and water column ammonia-N and (2) physiological response of V. natans to sediment Cu and water column ammonia-N.

Materials and Methods

Plant materials and culture

Vallisneria natans were collected from the downstream region (34°53′22″N, 113°41′13″E) of the Suoxu River, the tributary of the Huai River in Henan Province in central China. Uniform-sized plants (31.54 ± 6.32 cm tall, 5.03 ± 0.67 g fresh weight) were chosen for the experiment. The sediments used for the treatments were riverside soils containing 1.08% organic matter, 600 mg kg−1 total N, and 25.75 mg kg−1 Cu. The well water from the riverside contained 1.4 mg L−1 total N and 0.36 mg L−1 ammonia-N. In addition, nine large buckets (top diameter 84 cm, bottom diameter 67 cm, height 85 cm), 54 small basins (top diameter 12 cm, bottom diameter 8.7 cm, height 9.9 cm), and standard solutions of Cu (CuSO4⋅5H2O) and NH4+ solutions (NH4Cl) were used.

Treatments

Given that ammonia-N enrichment in the Huai River varied from 0.02 to 15.43 mg L−1 (Ren et al., 2015) and V. natans cannot survive levels higher than 8 mg L−1 (Zhu et al., 2015), we selected three levels of ammonia-N in this study (0, 3, and 6 mg L−1; LN, MN, and HN, respectively). In addition, three levels of Cu in sediment (control and Cu added at levels of 100 and 200 mg kg−1; LCu, MCu, and HCu, respectively) were selected based on the finding that the Cu sediment concentration in the polluted Huai River can reach 208.8 mg kg−1 (Yuan et al., 2015b). The three levels of ammonia-N content in the water column and three levels of Cu in the sediment produced 9 experimental treatments (LNLCu, LNMCu, LNHCu, MNLCu, MNMCu, MNHCu, HNLCu, HNMCu, and HNHCu). Each treatment was replicated three times.

The Cu treatments were created by adding a CuSO4 solution to the original soil samples. The treatments were made by taking a standard solution of Cu [CuSO4⋅5H2O] that had 9.7656 g CuSO4⋅5H2O and adding purified water to amount to 1 L in a volumetric flask (Cu2+ 2500 mg L−1). The low Cu treatment (LCu) consisted of 500 g soil, the medium Cu treatment (MCu) had 500 g soil with 20 ml CuSO4 solution added, and the high Cu treatment (HCu) was made by combining 40 ml CuSO4 solution with 500 g of soil. The three Cu concentration levels were calculated (LCu 25.75 ± 6.02 mg kg−1, MCu 125 ± 6.02 mg kg−1, HCu 225 ± 6.02 mg kg−1) according to the Cu concentration measured in the original soil (25.75 ± 6.02 mg kg−1). Each Cu treatment level had three ammonia-N concentrations (0, 3, and 6 mg L−1) that were created by adding a certain amount of NH4Cl solution.

The experiment began on 20 June 2014 and lasted for two weeks. Each small basin was filled with sediments and wrapped by plastic wrap, and then single plant was placed in each prepared basin through a little hole of the plastic wrap. The plastic wrap was used to avoid the removing of the Cu from sediments to water column. Three buckets were used for each sediment Cu level and nine buckets were utilized for the three levels of sediment Cu. Well water was added to each large bucket to create a water depth of 60 cm. Six small basins filled with the same level of sediment Cu were placed in one large bucket. Nine treatments were made by adding the three ammonia-N concentrations (0, 3, and 6 mg L−1) to the buckets with the three levels of sediment Cu. The buckets were then randomly positioned outside where there were no shade differences. The ammonia-N concentration of each large bucket was monitored daily and kept constant by adding an appropriate amount of NH4Cl solution. During the experiment, the concentration of ammonia-N in the water column ranged from 1.6 to 3.1 mg L−1 in the MN treatments, with an average of 2.4 mg L−1, and 4.2 to 6.1 mg L−1 in the HN treatments, with an average of 5.2 mg L−1. Water temperature was kept at 25.1–31.5 °C and underwater light intensity at noon ranged from 20,700 to 39,400 lux during the experimental period. The periphyton and phytoplankton in large bucket were removed through 100-mesh sieves every day. After the experiment, the Cu2+ concentration in the water from each large bucket was sampled and all levels were <0.01 mg L−1.

Harvest and chemical analysis

The plants were harvested after 1 and 2 weeks of treatments. Three small basins from each treatment were randomly selected for measurement at each harvest time. After harvest, the periphyton attached to the plant leaves was removed with a soft brush. The plants were washed with purified water, and dry with blotting paper carefully. A whole plant was weighed, above- and below-ground portions were separated, and the fresh weights were recorded. The leaves were placed in an ice bath to obtain the content of chlorophyll, NP-SH, and free proline. The biomass, chlorophyll, NP-SH, and free proline content were measured in the first harvest and only the biomass was measured in the second harvest. To measure the biomass, above- and below-ground portions were separated, and the fresh weights were recorded.

The Cu concentration of the original soil was measured using inductively coupled plasma atomic emission spectrometry (ICP-MS 7700x, Agilent Technologies, USA). The Cu2+ concentration of the water body in the large bucket was measured by plasma atomic emission spectrometry (ME-ICP02, ALS Minerals/ALS Chemex Co. Ltd, Guangzhou, China). The concentration of ammonia-N was measured with a HACH DR 2800 Spectrophotometer (HACH Company, Loveland, CO, USA). Relative growth rate (RGR) was calculated as RGR=InW2−InW1∕t, where W1 and W2 indicate the mean fresh weight at the first and second week, respectively, and t isthe growth time in days. In order to obtain the concentration of chlorophyll, 200 mg of sample was extracted using 25 ml 95% ethanol in the dark for 24 h at room temperature. The leaf chlorophyll concentration was measured by UV–vis spectroscopy, and the absorbance of the extracts was determined at 645 and 663 nm wavelengths (Lichtenthaler, 1987). The chlorophyll concentration was calculated using the equation described by (Arnon (1949); Su et al., 2012). Non protein thiol was determined following the method of Sedlak & Lindsay (1968). The molar extinction coefficient of 13,100 at 412 nm was used to estimate the thiol content and the values were expressed in nmol mg−1 of protein (Patra & Swarup, 2000). The free proline concentration was determined by the rapid colorimetric method described by (Bates, Waldren & Teare, 1973). The concentration of chlorophyll, NP-SH, and free proline of the leaves was calculated on the basis of fresh weight.

Data analysis

Growth indications (RGR and above- and below-ground biomass) and physiological indexes (chlorophyll, NP-SH, and free proline) were analyzed with one-way ANOVA. In addition, a two-way ANCOVA, with sediment Cu and ammonia-N as main factors and plant characteristics (RGR, above- and below-ground biomass, chlorophyll, NP-SH, and free proline) as covariates, was used to test the effects of sediment Cu and ammonia-N on plants. All statistical analyses were performed in SPSS19.0 software (SPSS, Chicago, IL, USA).

Results

Growth indicators of V. natans

Both ammonia-N and sediment Cu had significant effects on RGR (p < 0.01), but the influence of ammonia-N was more dramatic (p < 0.001, Table 1). The RGR was significantly lower in the MNLCu and HNLCu treatments than in the ammonia control (LNLCu) (Fig. 1). Compared to the Cu control (LNLCu), RGR was significantly decreased in the LNMCu and LNHCu treatments (Fig. 1). Moreover, the highest ammonia-N and Cu combination (HNHCu) resulted in the lowest value of RGR.

Table 1 Two-way ANOVA results (F value) for the relative growth rate (RGR), above-ground biomass, below-ground biomass, chlorophyll, non-protein thiol (NP-SH), and free proline of Vallisneria natans using water column ammonia-N and sediment Cu as dependent variables.

Dependent variable	Ammonia-N	Sediment Cu	Ammonia-N × Sediment Cu	
RGR	112.67∗∗∗	7.70∗	0.48ns	
Above-ground biomass	109.51∗∗∗	0.68ns	2.32ns	
Below-ground biomass	0.98ns	4.38∗	0.15ns	
Chlorophyll	9.86∗∗∗	4.27∗	0.20ns	
NP-SH	25.80∗∗∗	5.24∗	0.86ns	
Free proline	16.95∗∗∗	1.34ns	0.23ns	
Notes.

Statistical significance indicated through asterisk(s): ∗∗∗p < 0.001, ∗∗p < 0.01, ∗p < 0.05, ns p > 0.05.

Figure 1 Relative growth rates (RGR) of nine treatments across two harvest times of Vallisneria natans.

Mean and standard errors of three replicates were shown; different letters represent significant difference at p < 0.05 between treatments. Abbreviations are the same to those shown in Table 2.

The above-ground biomass differed significantly among the ammonia-N treatments (p < 0.001), whereas the below-ground biomass showed significant variation among the sediment Cu groups (p < 0.05, Table 1). The below-ground biomass decreased significantly with increasing Cu sediment levels when water column ammonia-N levels were constant (Fig. 2A). Similarly, the above-ground biomass also decreased with an increasing water column ammonia-N concentration when the sediment Cu levels were stable (Fig. 2B).

Figure 2 Biomass of nine treatments at the first harvest.

(A) Below-ground biomass (g) and (B) Above-ground biomass (g) of Vallisneria natans of nine treatments from the first harvest. Mean and standard errors of three replicates were shown; different letters represent significant difference at p < 0.05 between treatments. Abbreviations are the same to those shown in Table 2.

Physiological indexes of V. natans

Chlorophyll

Both ammonia-N and sediment Cu had significant effects on chlorophyll (p < 0.05), but the influence of ammonia-N was more dramatic (p < 0.001, Table 1). Increasing levels of both water column ammonia-N and sediment Cu levels led to lower chlorophyll concentrations. The lowest chlorophyll content corresponded to the treatment with the highest ammonia-N and sediment Cu levels (HNHCu, Fig. 3).

Figure 3 Chlorophyll levels of nine treatments at the first harvest.

Chlorophyll concentrations (mg g−1) of Vallisneria natans at nine different treatments were obtained during the first harvest. Mean and standard errors of three replicates were shown; different letters represent significant difference at p < 0.05 between treatments. Abbreviations are the same to those shown in Table 2.

NP-SH

Compared to the ammonia control (LNLCu), NP-SH significantly increased in the medium and high ammonia-N treatments (MNLCu and HNLCu, Table 2). In addition, when compared to the Cu control (LNLCu), NP-SH significantly increased in the medium and high Cu treatments (LNMCu and LNHCu, Table 2). The NP-SH concentration was more sensitive to ammonia-N (p < 0.001) than to sediment Cu (p < 0.05, Table 1), although it was significantly affected by both factors.

Table 2 Non-protein thiol (NP-SH) and free proline contents of Vallisneria natans in nine treatments obtained from the first harvest.

Treatment	LNLCu	LNMCu	LNHCu	MNLCu	MNMCu	MNHCu	HNLCu	HNMCu	HNHCu	
NP-SH (µmol g−1)	3.50 ± 0.61d	4.16 ± 0.31cd	4.44 ± 0.85bc	4.55 ± 0.18bc	5.26 ± 0.46ab	5.53 ± 0.19a	5.47 ± 0.48a	5.59 ± 0.42a	5.60 ± 0.21a	
Free proline (µg g−1)	23.28 ± 3.69c	25.65 ± 4.69c	41.02 ± 27.56bc	54.03 ± 9.88ab	62.2 ± 15.91ab	62.31 ± 8.38ab	64.68 ± 10.79ab	71.77 ± 15.97a	73.71 ± 21.77a	
Notes.

AbbreviationsL Low

M Medium

H High

N Nitrogen

Cu Copper, LN (0 mg N L−1), MN (3 mg N L−1), HN (6 mg N L−1), LCu (25.75 mg Cu kg−1), MCu (125.75 mg Cu kg−1), HCu (225.75 mg Cu kg−1)

Mean and standard error of three replicates is shown; different letters represent significant difference at p < 0.05.

Free proline

Free proline levels were primarily determined by ammonia-N (p < 0.001), whereas the impact of sediment Cu was negligible (p > 0.05, Table 1). The free proline content in the medium and high ammonia-N treatments (MNLCu and HNLCu) was significantly higher than in the ammonia control (LNLCu, Table 2). Sediment Cu level showed little effect on the free proline content, and the higher sediment Cu did not produce a greater free proline content when the ammonia-N content was constant (Table 2).

Discussion

A reduction in RGR and biomass of submerged plants has been reported under high ammonia-N (Wang et al., 2008; Zhang et al., 2013) and Cu concentrations (Srivastava et al., 2006; Xue et al., 2010). Our study shows that the below-ground biomass of V. natans decreased with increasing Cu sediment levels, indicating that excessive sediment Cu (heavy metal toxicity) could result in significant damage to the root of V. natans. This injury, which can include stunted roots and poor growth initiation, might result in low water and nutrient uptake and a disturbance in metabolism (Påhlsson, 1989). In contrast, above-ground biomass significantly decreased with increasing ammonia-N concentrations, indicating that excessive water ammonia-N can cause significant toxicity to the leaf of V. natans. Previous studies showed that ammonia-N was the preferred N-source for submerged plants to uptake N through the shoots (Cedergreen & Madsen, 2003; Racchetti et al., 2010). In summary, our experiment showed that above-ground biomass was affected by water ammonia-N and below-ground biomass by sediment Cu. Thus, the RGR of V. natans was more affected by water ammonia-N than sediment Cu, indicating that the more dramatic leaf as compared to root reduction may result in the decrease of RGR.

Several studies have demonstrated a reduction in chlorophyll content under Cu stress in a variety of aquatic plants, including Potamogeton pusillus (Monferrán et al., 2009), Elodea canadensis (Malec et al., 2009), and Lemna sp. (Geoffroy, Frankart & Eullaffroy, 2004). The present study showed a similar trend, with a clear reduction in the chlorophyll content of V. natans after exposure to sediment Cu, thus supporting the findings of earlier studies on submerged plants (e.g., Geoffroy, Frankart & Eullaffroy, 2004; Malec et al., 2009). Damage from sediment Cu likely results from high Cu contents distorting chlorophyll structure and thereby inhibiting the synthesis of photosynthetic pigment (Prasad et al., 2001). Previous studies showed that excessive ammonia decreases total chlorophyll in aquatic plants such as Myriophyllum (Saunkaew, Wangpakapattanawong & Jampeetong, 2011) and Egeria densa (Su et al., 2012). Our results support these early findings and demonstrate that total chlorophyll in V. natans was also reduced by high ammonia-N. Ammonia-N appears to affect total chlorophyll in aquatic plants by damaging the photosynthetic system and inhibiting photosynthesis (Wang et al., 2008).

NP-SH, a class of low-molecular-weight-SH compounds, has been considered as an important plant defense source in response to heavy metals, including Cu (Maserti et al., 2005). Our results suggest that NP-SH content increased with rising sediment Cu levels. These results are supported by previous studies in which NP-SH significantly increased with higher sediment Cu levels (Morelli & Scarano, 2004; Srivastava et al., 2006; Fernández et al., 2014). NP-SH may induce resistance to heavy metals by protecting labile macromolecules against attack by the formation of free radicals in metabolic reactions and its oxidative stress (Patra, Swarup & Dwivedi, 2001; Mishra et al., 2006). Like sediment Cu, excess ammonia-N also leads to NP-SH accumulation in V. natans. The present results are in agreement with previous studies in which ammonia-N induced an increase of NP-SH content in submerged plants (Wang et al., 2010a). Therefore, the accumulation of the NP-SH likely indicates that plants are being stressed by sediment Cu and water ammonia-N.

Proline is a common free amino acid in plant tissues that contributes to osmotic adjustment, detoxification of reactive oxygen species, and protection of membrane integrity (Sharma & Dietz, 2006; Ashraf & Foolad, 2007). The accumulation of proline under heavy metal stress conditions has been reported in aquatic macrophytes such as Salvinia natans (Mohan & Hosetti, 2006), Lemna gibba (Megateli, Semsari & Couderchet, 2009), and Najas indica (Singh et al., 2010). Our results also indicate that proline levels increased in V. natans in response to excess sediment Cu. Proline-Cu complexes likely enhance the tolerance of plants to heavy metals by reducing free metal ion activities through the formation of metal–proline complexes (Xiong, Liu & Geng, 2006; Szabados & Savouré, 2010). Proline accumulation can also increase dramatically in response to rising ammonia concentrations in aquatic environments (Xu et al., 2012; Lee et al., 2013). We also found that proline accumulated in V. natans under high water column ammonia-N conditions. Previous studies revealed that excess ammonia-N in plant tissues caused cellular and whole plant water imbalance by decreasing Ca2+ and K+ uptake (Britto & Kronzucker, 2002; Roosta & Schjoerring, 2007). The accumulation of proline may prevent water loss by sustaining cell turgor, maintaining membrane integrity, and inhibiting protein denaturation (Hong et al., 2000; Kim et al., 2004; Neuberg et al., 2010).

Most growth indicators and physiological indexes of V. natans are significantly correlated with the concentration of water column ammonia-N and sediment Cu (p < 0.05). We also found that ammonia-N concentration (p < 0.001) played a more crucial role in affecting plant growth indicators and physiological indexes than sediment Cu (p < 0.05). Moderate to high levels of sediment Cu enhanced the toxicity of water column ammonia-N, and even moderate ammonia-N content yielded negative RGR when exposed to moderate levels of sediment Cu.

This study provides new and important insights into potential methods of ecological restoration after an environment has been damaged by heavy metals. Experiments assessing ammonia-N stress on V. natans showed that high ammonia-N content (>8 mg L−1) in the water column lead to severe plant damage (Zhu et al., 2015). Compared to previous studies, our experiments evaluated relatively lower ammonia-N content in the water column and further narrowed the tolerant water ammonia-N content to <6 mg L−1. Moreover, this study suggests that cross effect of various factors are non-neglectful, even this cross effect seems less significant. Moderate to high sediment Cu levels intensify ammonia-N stress on submerged plants and yield much lower tolerant water ammonia-N content (<3 mg L−1) for V. natans.

Supplemental Information

Data S1 Raw data

Click here for additional data file.

Additional Information and Declarations

Competing Interests

Author Contributions

Data Availability

The authors declare there are no competing interests.

Zhengjie Zhu conceived and designed the experiments, performed the experiments, analyzed the data, contributed reagents/materials/analysis tools, wrote the paper, prepared figures and/or tables.

Siyuan Song and Pengshan Li performed the experiments.

Nasreen Jeelani and Penghe Wang analyzed the data.

Hezhong Yuan and Jinghan Zhang prepared figures and/or tables.

Shuqing An and Xin Leng conceived and designed the experiments, reviewed drafts of the paper.

The following information was supplied regarding data availability:

The raw data has been supplied as Data S1.

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
