# Peer review of "Growth and physiological responses of submerged plant Vallisneria natans to water column ammonia nitrogen and sediment copper"

_PeerJ, doi:10.7717/peerj.1953_

## Round 0.1 · original submission · Minor Revisions

Kindly submit your revised manuscript as per the suggestions made by the reviewers.

·

Basic reporting

The manuscript involves the subject of water-column ammonia-nitrogen and sediment copper toxicity on the aquatic plant species Vallisneria natans. The work involves a mesocosm-like study where plants exposed to various combinations of treatments are compared over a two week period of growth in which samples are extracted in two time periods (1 week and 2 week) for comparison. Various parameters are analyzed to determine significance of treatments.

Basic Reporting: The manuscript is very well written and the experiment is clear and unambiguous. The work appears to be sound and results are valid. Introduction and background information are thorough and supports the justification for the work performed. Tables and figures are properly supported and the article provides appropriate information to stand alone in its conclusions. The raw data supports the work. A couple of specific observation in the introduction: Lines 44-46 states that vegetation restoration has emerged as an effective way of improving water quality by “controlling eutrophication”. Submerged aquatic vegetation is often negatively impacted by eutrophication, particularly seagrasses. Might stand to be better supported by explaining a little better the context and mechanism by which this is supported.
Also, in this same introductory paragraph it is stated in lines 48-49 that “successful ecological restoration depends on planting suitable submerged macrophytes in a sustainable condition such as water ammonia-N and sediment-Cu. Sentence structure is awkward. Ammonia-N and sediment-Cu are not “sustainable conditions”.

Experimental design

Experimental Design: The experimental design is sound and properly addresses the research question. The authors have done a good job of keeping the design simple and straight-forward allowing for direct results without confounding variables. The parameters selected appear to be strong representations of accurate plant response. There are a couple of clarifications that need to be addressed.
Lines 114-115 – the plants were described as being wrapped in plastic wrap. This is very unclear. Why were they wrapped in plastic wrap? Where they not planted into soil. Also, were the plants collected as bare root specimens and then planted into the experimental soil media. Were they planted as single rosets? Please clarify.
Lines 134-135 describe biomass measurements as fresh weights. Need more specific explanation. How were these handled and treated? Were they patted dry and free of all water? Why not dry weights? How could the authors be sure these weights were not tainted by surface water?

Validity of the findings

Validity of the Findings: The results appear to be strongly supported by the analysis and therefore the results are validated. The stats are in order and subject to good control. The conclusions are sound and there is little to no speculation. The data and analysis is well supported by appropriate citations. This work should contribute very well to the body of work related to submerged aquatic vegetation ecology, especially as it pertains to N and Cu loadings. The information in new and yet consistent with previous related works. The experiment is readily reproducible.

Additional comments

I find this work to be very well done, the conclusions are sound and the information is highly relevant. My compliments on a job well done.

Reviewer 2 ·

Basic reporting

This study aimed to examine the combined effects of water column ammonia-N contents and sediment copper treatments on the growth and physiological responses of a submersed macrophyte Vallisneria natans, using relative growth rate, above- and below- ground biomass, and some biochemical parameters. This paper emphasized the influence of sediment copper on the ammonia toxicity and tested that high levels of copper would significantly affect the tolerant values of ammonia-N for Vallisneria natans. In addition, this paper is well organized and the results were given a good interpretation. Therefore, I recommend its acceptance for publication in this journal after minor revisions.

Experimental design

the experiment is a normal one, and experimental design is right.

Validity of the findings

this study found that high levels of copper would significantly affect the tolerant values of ammonia-N for Vallisneria natans, which is important for ecological resotration of this species in water doby.

Additional comments

The authors did an interesting work. This study aimed to examine the combined effects of water column ammonia-N contents and sediment copper treatments on the growth and physiological responses of a submersed macrophyte Vallisneria natans, using relative growth rate, above- and below- ground biomass, and some biochemical parameters. This paper emphasized the influence of sediment copper on the ammonia toxicity and tested that high levels of copper would significantly affect the tolerant values of ammonia-N for Vallisneria natans. In addition, this paper is well organized and the results were given a good interpretation. Therefore, I recommend its acceptance for publication in this journal after minor revisions.

Specific comments:
(1) Line 58 “at high levels, it can be toxic” can be changed as “it can be toxic at high levels”.
(2) Line 118-120 This paper has described the light conditions and water temperatures during the treatments, but what about the periphyton and phytoplankton biomass in the system? This could also be a major factor influencing the growth of macrophyte.
(3) Line 117-118 Please keep the same digits, and they can be modified as follows: with an average of 2.4 mg L-1…. with an average of 5.2 mg L-1…
(4) Line 190-192 Please change can and may into could and might.
(5) Line 213 It should be “has been considered as…”
(6) Line 241 change plays into played
(7) Line 243 change enhance into enhanced
(8) Line 247 change leads into lead
(9) Fig.1, 2, and 3 Please change 0, 3, 6 (ammonia-N contents) into LN, MN, and HN in figures. In addition, all figure captions should keep constant and please add abbreviations of ammonia-N contents.
(10) Table 2 abbreviations of contents of ammonia-N and copper for different treatments should be denoted as figure captions, not merely low, medium and high.

---

## Round 0.2 · accepted · Accept

I have evaluated the revision and your rebuttal and the manuscript is up to the standard of the journal. I recommend the revised article for publication in PeerJ.